# Modelling contrail cirrus using a double-moment cloud microphysics scheme in the UK Met Office Unified Model

Weiyu Zhang<sup>1</sup>, Paul R. Field<sup>1,2</sup>, Kwinten Van Weverberg<sup>3,4</sup>, Piers M. Forster<sup>1</sup>, Cyril J. Morcrette<sup>2,5</sup>, Alexandru Rap<sup>1</sup>

<sup>1</sup>School of Earth and Environment, University of Leeds, Leeds, LS2 9JT, UK

<sup>2</sup>Met Office, Exeter, EX1 3PB, UK

10

<sup>3</sup>Department of Geography, Ghent University, Ghent, Belgium

<sup>4</sup>Royal Meteorological Institute of Belgium, Brussels, Belgium

<sup>5</sup>Department of Mathematics and Statistics, Exeter University, Exeter, EX4 4QE, UK

Correspondence to: Alexandru Rap (a.rap@leeds.ac.uk)

Abstract. Contrail cirrus is the largest contributor to aviation effective radiative forcing (ERF), but remains highly uncertain (~70%). Recent research has highlighted the critical role of cloud microphysical schemes in contrail cirrus climate modelling. In this study, we implement a contrail parameterisation in the double-moment cloud microphysics scheme, Cloud AeroSol Interacting Microphysics (CASIM), within the regional configuration of the UK Met Office Unified Model (UM). We first investigate a contrail cluster model experiment, showing that the simulated contrails retain a high ice crystal number concentration for several hours before declining. Ice water content increases during the early stage of the lifecycle before gradually decreasing. In addition, as the contrail cluster gradually sediments below flight levels, there is an increase in both contrail ice number concentration and water content. We also perform regional simulations over a European domain, estimating a regional annual mean contrail cirrus ERF of 0.93 W m<sup>-2</sup>, within the range of previous climate modelling estimates. Using a range of initial contrail width, depth and ice crystal size based on contrail observations, we estimate an annual mean European regional contrail cirrus ERF range of 0.19 W m<sup>-2</sup> to 2.80 W m<sup>-2</sup>. The seasonal cycle of contrail cirrus ERF is mainly driven by the background meteorology and the natural clouds vertical structure. Our study highlights the critical need for double-moment cloud microphysics in global climate models to realistically simulate contrail microphysical properties, contrail lifetime, and climate impact.

#### 1 Introduction

Currently, aviation contributes approximately 3.5% to overall anthropogenic radiative forcing (RF), driven by CO<sub>2</sub> emissions but more importantly, by non-CO<sub>2</sub> effects (Lee et al., 2021). About two-thirds of aviation's climate impact is attributed to these non-CO<sub>2</sub> effects, including emissions of nitrogen oxides (NO<sub>x</sub>), water vapour, and aerosols, as well as the formation of contrails. Among these, contrails and contrail cirrus (the cirrus clouds that evolve from contrails) represent the largest contribution to

aviation radiative forcing. Improving our understanding of contrail cirrus radiative forcing is crucial for assessing the climate impact of aviation. As global air travel demand is expected to double in the coming decades (Dray et al., 2022; Gössling and Humpe, 2020), accurately estimating the climate effects of contrail cirrus is increasingly crucial for developing effective mitigation strategies and guiding technological and operational advancements.

Contrails form when the temperature drops during the mixing of hot aircraft exhaust with cold ambient air, inducing supersaturation with respect to liquid water. When the temperature falls below the critical threshold for contrail formation, the liquid droplets form ice crystals via homogeneous or heterogeneous freezing. In ice-supersaturated air, contrails can grow in size and ice water content and evolve into contrail cirrus, which may interact with pre-existing natural cirrus clouds (Kärcher, 2018). During the day, contrails scatter incoming shortwave solar radiation back into space, creating a cooling effect. At all times, they also trap terrestrial radiation, which lowers outgoing longwave infrared radiation and produces a warming effect. Overall, contrail cirrus typically results in a net warming effect on the climate (Zhang et al., 2025; Bickel et al., 2020; Teoh et al., 2024; Singh et al., 2024).

The climate impact of contrail cirrus is driven by its microphysical properties and coverage, which vary with contrail age and ambient meteorological conditions. Newly formed contrails are characterized by their high ice crystal number concentrations, which can be orders of magnitude greater than those in natural cirrus clouds, often exceeding 10<sup>4</sup> cm<sup>-3</sup> compared to typical natural cirrus values of 10-100 cm<sup>-3</sup> (Schumann et al., 2017; Wang et al., 2023). Contrail ice crystals grow through vapour deposition and aggregation, while their subsequent evolution is strongly influenced by ambient conditions, including ice supersaturation, vertical wind shear, and turbulent mixing processes (Lewellen, 2014; Gruber et al., 2018). As contrails age, ice crystal number concentrations decrease due to dilution and aggregation, gradually resembling those of natural cirrus (Unterstrasser and Gierens, 2010). When contrails form in regions where natural cirrus clouds are present, complex interactions occur as they compete for available water vapour, leading to modified growth rates of contrail and natural ice crystals (Bock and Burkhardt, 2016b). This microphysical evolution and interaction with the surrounding environment and cloud determine both the lifetime and radiative properties of contrails, ultimately shaping their climate impact (Kärcher, 2018).

The evolution of contrail microphysical properties can be simulated in great detail using high-resolution large eddy simulation (LES) models and parameterized Lagrangian models (e.g. Lewellen (2014), Unterstrasser (2016), Schumann (2012), and Lottermoser and Unterstrasser (2025)), while numerical weather prediction and climate models capture large-scale properties and rapid atmospheric adjustments (e.g. Chen and Gettelman (2013) and Bock and Burkhardt (2016a)). Climate models provide a broader scale representation of contrail evolution, accounting for changes in synoptic conditions and interactions and overlap with natural clouds, which significantly influence the contrail lifecycle and are therefore crucial for accurately estimating contrail cirrus effective radiative forcing (ERF).

Despite substantial progress in contrail modelling, important uncertainties of approximately 70% remain in the climate forcing estimates of contrail cirrus, as reported in the latest aviation climate assessment of the Intergovernmental Panel on Climate Change (IPCC) Sixth Assessment Report (AR6) (Szopa et al., 2021). The main sources of uncertainty are associated with upper-tropospheric humidity and ice supersaturation variability, the treatment of contrail cirrus and its interactions with natural

clouds, and the radiative transfer response to contrail cirrus, with additional unquantified uncertainties from contrails forming within natural clouds and the effects of soot aerosols in contrail cirrus ice crystals on radiative transfer. Moreover, the assessment of the uncertainty is hindered by the limited number of available independent climate models. There are currently only a couple of climate models providing independent contrail cirrus ERF estimates, one based on the ECHAM (Bickel et al., 2020; Bock and Burkhardt, 2016a) and another one based on CAM (Chen and Gettelman, 2013; Gettelman et al., 2021) global climate models. They have different cloud microphysical parameterisations in general and contrail cirrus parameterisations in particular. ECHAM represents contrail cirrus as a separate cloud species, while CAM merges contrail ice into other ice clouds. Recently, in Zhang et al. (2025) we implemented a contrail cirrus parameterization in the global configuration of the UK Met Office Unified Model (UM). However, in that work it was not possible to provide a new fully independent contrail cirrus ERF estimate due to the UM model's one-moment cloud microphysics scheme which poorly represents the small ice crystal size typical of contrail cirrus. This highlighted the need for more advanced bulk cloud microphysics representations, such as double-moment schemes, which can better capture contrail ice crystal number concentrations and size variability.

In this study, we implement a contrail parameterisation in the double-moment cloud microphysics scheme, Cloud AeroSol Interacting Microphysics (CASIM), within the regional configuration of the UM with a horizontal grid resolution of approximately 4 km. This enables the UM to represent the high ice-particle number concentration of young contrails, which is critical for the simulation of contrail cirrus evolution and climate impacts. We evaluate the simulated contrail cirrus evolution in CASIM-UM through a contrail cluster experiment by comparing with observations and results from other models, investigating its ability to capture the evolution of key contrail characteristics. Additionally, we perform a series of regional simulations over a European domain to estimate regional contrail cirrus ERF and compare with existing results from other models.

## 2 Methodology

85

95

#### 2.1 CASIM cloud microphysics

A contrail scheme (Chen et al., 2012; Zhang et al., 2025) is implemented in the CASIM cloud microphysics parameterisation within the UM in this study. CASIM is developed to represent cloud microphysical processes and aerosol–cloud interactions over varying spatial and temporal scales. It has recently been fully coupled to the UM as described in Field et al. (2023). CASIM represents mixing ratios of mass and number concentration for five hydrometeor species: cloud liquid, cloud ice, rain, snow, and graupel. For each cloud species, the particle size distribution (PSD) is described using a generalised gamma function with constant shape parameters. The frozen cloud fraction is determined using a diagnostic relationship obtained for the mass and fraction of liquid cloud, applied to the total mass of cloud ice and snow. This approach assumes a Gaussian variability at subgrid scales, where variances are derived from turbulent properties, while bimodal distributions near entrainment zones are also considered (Van Weverberg et al., 2021). Radiative transfer calculations are performed by the Suite of Community Radiative Transfer Codes Based on Edwards and Slingo (SOCRATES) (Manners, 2018; Edwards and Slingo, 1996),

incorporating six shortwave and nine longwave spectral bands. Cloud ice crystals are handled following Baran et al. (2016) within the radiation scheme, with a maximum–random overlap assumption for vertical cloud layers.

## 100 2.2 Contrail parameterisation

The contrail parameterisation implemented in CASIM is described in Chen et al. (2012) in detail and has been applied for contrail simulations in both the UM (Zhang et al., 2025) and CAM (Chen and Gettelman, 2013; Chen et al., 2012; Gettelman et al., 2021). Contrail formation in this parameterisation is triggered when the ambient temperature satisfy Schmidt-Appleman criterion (Appleman, 1953; Schumann, 1996). Contrails persist in ice supersaturated areas. According to Schumann et al. (1996), the critical temperature ( $T_c$ , °C) of contrail formation is defined by the empirical equation:

$$T_c = -46.46 + 9.43 \ln(G - 0.053) + 0.72 [\ln(G - 0.053)]^2, \tag{1}$$

with G (Pa  $K^{-1}$ ) given by

105

120

$$G = \frac{EI_{H_2O} \cdot c_p p}{\varepsilon Q(1-\eta)},\tag{2}$$

where  $EI_{H_2O}$  denotes the emission index of water vapour  $(kg_{H_2O} kg_{fuel}^{-1})$ ,  $c_p$  represents the specific heat of air at constant pressure  $(J kg_{air}^{-1} K^{-1})$ , p is the atmospheric pressure (Pa),  $\varepsilon$  is the molecular masses ratio of water to air  $((kg_{H_2O} mol^{-1}) (kg_{air} mol^{-1})^{-1})$ , Q refers to the specific combustion heat  $(J kg_{fuel}^{-1})$ , and  $\eta$  is the jet engine propulsion efficiency.

The critical relative humidity for contrail formation  $RH_c$  varies with G,  $T_c$  and the ambient temperature T, as described in Ponater et al. (2002) and Rap et al. (2010):

$$RH_c(T) = \frac{G \cdot (T - T_c) + e_{sat}^L(T_c)}{e_{sat}^L(T)},$$
 (3)

where  $e_{sat}^{L}$  represents the saturation vapour pressure over liquid water.

Contrail persistence requires the ambient air be to under ice supersaturation condition. Contrail ice mass and number concentration are introduced to cloud fields as increments. The ice mass is calculated from both aircraft engine-emitted water vapour and depositional water (the excess water vapour above ice supersaturation) within the contrail volume. The contrail volume is determined by multiplying the flight path distance d (m) by the cross section area C (m<sup>2</sup>). The contrail ice mass mixing ratio is obtained by

$$M = q_t \Delta t + \frac{d \cdot C}{V} \left( x - x_{sat}^i \right), \tag{4}$$

where  $q_t$  denotes the tendency of the aviation water vapour emission mixing ratio (kg kg<sup>-1</sup> s<sup>-1</sup>), defined as the mass of aircraft water vapour emission to the mass of dry air; V is the grid cell volume (m<sup>3</sup>); x refers to the ambient specific humidity (kg kg<sup>-1</sup>), i.e., the ratio of the mass of aviation water vapour emission to the total air mass; and  $x_{sat}^i$  is the ice saturation specific humidity

(kg kg $^{-1}$ ) under the ambient temperature and pressure. In this study, the cross section area C is calculated assuming an elliptical shape, which resembles the cross section of 5 minute old contrails (Paoli and Shariff, 2016).

The initial contrail number concentration is derived under the assumption of spherical particles with a prescribed initial ice crystal radius and density. The impact of contrails on cloud fraction in CASIM is diagnosed by the bimodal cloud fraction scheme described by Van Weverberg et al. (2021). Once formed, contrail ice is integrated into the model's hydrological cycle, allowing it to persist, evolve, and evaporate over time, with lifetimes ranging from several minutes to hours. This parameterisation enables the model to simulate both young contrails (with high concentrations and small sizes relative to the background natural cirrus) and contrail cirrus that evolves from them. Additionally, the model captures their interactions with the ambient environment and existing clouds.

# 135 **2.3 Model setup**






The CASIM-UM model simulations in this study were conducted over a European domain covering 35°N-58°N and 10°W-22°E as illustrated by Fig. 1, an area of intense air traffic. The domain is defined on a rotated-pole grid, which has a horizontal resolution of ~0.04° and a size of 500 × 600 grid points. This European domain is nested within a global simulation using the Global Atmosphere 7.2/Global Land 8.1 configuration of the UM (Walters et al., 2019). CASIM-UM employs 70 vertical levels, reaching up to 40 km, with a vertical resolution of approximately 500 m in the upper troposphere and lower stratosphere (UTLS). The model time step is one minute. The lateral boundary conditions of the regional domain are updated on an hourly basis. We perform two sets of contrail simulations with CASIM-UM: an idealised contrail cluster experiment and a European regional contrail cirrus simulation.

For the contrail cluster experiment, a contrail cluster is introduced over a small region in southern Spain, centred at 5.0°W, 37.5°N (marked as the purple dot in Fig. 1), at altitudes between 170 and 250 hPa. The contrail cluster is initialized over a 30-minute period from 00:30 to 01:00 UTC on 5 December 2018, a time period characterized by ice supersaturation in the UTLS and clear-sky conditions. This setup enables the analysis of the temporal evolution of persistent contrail microphysical properties.

For the regional contrail simulation over a European domain (outlined in red in Fig. 1), contrails are initialized across the entire domain at every model time step, based on the Aviation Environmental Design Tool (AEDT) air traffic inventory (Wilkerson et al., 2010). Due to computational constraints, the simulations are run for the first 10 days for each month of 2018, with initial conditions taken from the global operational Met Office analysis, allowing for the estimation of annual contrail cirrus properties. In all simulations, contrail increments are determined using the contrail parameterisation with the AEDT air traffic inventory, which provides gridbox-aggregated distance flown and water vapour emission for 2006 (Wilkerson et al., 2010). To reflect the growth in air traffic from 2006 to 2018, a scaling factor of 1.58 is applied to the AEDT inventory informed by the increase in total flight distance, following Lee et al. (2021). Each simulation is performed in pairs (one with contrails and one without) to quantify the contrails influence on cloud microphysics and radiation.

Figure 1: Flight distance in km year<sup>-1</sup> from AEDT air traffic inventory at 197 hPa over the European domain considered in this study. The red box outlines the CASIM-UM European domain and the blue dot marks the location of the contrail cluster in the idealised contrail cluster experiment.

Observations from the Contrail Library (COLI; (Schumann et al., 2017)) are used to evaluate the contrail microphysical properties derived in our contrail cluster experiments and identify the range of initial contrail properties. COLI comprises observations of 230 individual contrails from flight campaigns and remote-sensing measurements dating back to 1972. It systematically characterizes contrail-cirrus microphysics for plume ages ranging from a few seconds to 11.5 hours, at flight altitudes of 7.4–18.7 km and ambient temperatures between –88 and –31 °C. For each entry, COLI provides mean values (with uncertainty bounds) of key parameters—ice-particle number concentration, ice-water content, optical depth, and plume width—along with metadata on aircraft type, engine, fuel, atmospheric conditions, instrumentation, and literature references. This dataset has become the standard benchmark for validating contrail-cirrus models and is widely cited in subsequent studies. There is a large variability in observations of young contrail ice volume and ice crystal size due to variations in aircraft engine design, fuel composition, and ambient atmospheric conditions (Schumann et al., 2017). Simulations of contrail cirrus ERF are sensitive to these parameters (Chen et al., 2012; Zhang et al., 2025; Lee et al., 2021). To investigate the sensitivity of our model to these initial contrail parameters, we consider the range of values based on the COLI observations (Table 1). These represent the range of potential young contrails properties at five minutes after formation (Fig. 2), well after the vortex phase,

and provide a basis for estimating the range of possible contrail cirrus ERF values. The best estimates within these ranges are based on linear regressions of the observations for contrail age younger than ten minutes (Fig. 2). For both the contrail cluster experiment and regional simulations over the European domain, we apply three sets of initial contrail parameters, as shown in Table 1. These represent the range of potential microphysical properties of young contrails at five minutes after formation (Fig. 2), well after the vortex phase, and provide a basis for estimating the range of possible contrail cirrus ERF values.

Table 1. Initial contrail microphysical parameter sets used in CASIM contrail simulations. The table presents values for contrail ice crystal radius ( $\mu$ m), contrail volume width (m), and contrail volume depth (m) for three simulations.

| Parameter set | Contrail ice radius (µm) | Contrail volume width (m) | Contrail volume depth (m) |
|---------------|--------------------------|---------------------------|---------------------------|
| Upper bound   | 2.0                      | 300                       | 300                       |
| Best estimate | 3.0                      | 250                       | 200                       |
| Lower bound   | 4.0                      | 200                       | 150                       |

Figure 2: Scatterplots of contrail microphysical properties from contrail age younger than ten minutes, derived from the COLI library. (a) Contrail volume width, (b) contrail volume depth, and (c) contrail particle radius. The red line indicates the best estimate for each parameter corresponding to a 5-minutes old contrail, with the grey dashed lines indicating the upper and lower bounds. The blue lines are the linear regression lines.

#### 3 Results




## 3.1 Contrail cluster experiment

The impact of contrail cirrus on cloud microphysical properties is analysed by comparing simulations with and without contrails. In the contrail cluster experiment, feedback from natural clouds is negligible, as the selected area on the experiment

day was chosen to be under clear sky conditions. Additionally, there is large-scale ice supersaturation that allows contrails to persist and evolve over a long period. The simulated contrail properties are averaged over the contrail cluster and compared with observations from COLI (Schumann et al., 2017), a comprehensive database of contrail measurements spanning decades, multiple locations, various aircraft types, and a wide range of meteorological conditions. This extensive dataset has enabled the systematic cataloguing of how contrail properties evolve as a function of their age. To identify contrail cirrus gridboxes, a threshold of 0.001 for cloud fraction change was applied. We focus on analysing contrails up to 4 hours old, a point near which contrail cirrus properties begin to resemble those of natural cirrus (Bock and Burkhardt, 2016b).

Figure 3 shows the background relative humidity with respect to ice (RH<sub>ice</sub>) over the area where the contrail cluster is initially injected. In the UTLS, roughly between 210 and 270 hPa, there exists an ice supersaturated layer where RH<sub>ice</sub> exceeds 100%, reaching approximately 130% at 250 hPa. This ice supersaturation indicates conditions that support the formation and persistence of contrails. Meanwhile, around the levels between 300 and 350 hPa, the air is subsaturated. Since the contrail cluster is initially injected between 170 and 260 hPa, it may dissipate as sediments into these drier layers below.





Figure 3: Vertical profile of relative humidity with respect to ice ( $RH_{ice}$ ). Red horizontal dashed lines at ~210 hPa and ~270 hPa indicate the boundaries of ice supersaturation, i.e.  $RH_{ice} > 1.0$  (vertical dashed line).

A high ice number concentration is initiated at contrail formation (Fig. 4a), which then decreases due the conversion to snow and dilution from horizontal mixing, dropping by over an order of magnitude within four hours. The results show good agreement with observations, capturing a similar decline and range of variation over this period, although the simulated initial in-contrail number concentration is smaller than observed by 0.65 cm<sup>-3</sup> on average. This is likely due to the overestimation of the initial contrail fraction, since in-contrail values are computed in the model as the change in ice water content divided by the change in cloud fraction. In our study, contrail fraction is defined as the difference in total cloud fraction between a

simulation with contrails and one without, and it is treated in the same way as natural cloud ice. In CASIM, contrail fraction is diagnosed using the bimodal cloud fraction scheme (Van Weverberg et al., 2021), which follows the approach of Abel et al. (2017) to diagnose the ice cloud fraction. It incorporates the additional contrail ice into the grid-box subgrid distribution of humidity. As a result, contrail moisture is mixed throughout the grid-box after a single time step—much faster than in reality—leading to lower in-contrail values than observed. In addition, the conversion of cloud ice mass into cloud fraction depends on the assumed mass distribution. In this work, the initial contrail volume width is about 200 m, much smaller than the grid-box area (4 km by 4 km), introducing an overestimation to the initial contrail fraction. Higher-resolution simulations with horizontal grid spacing of around 200 m would be required to overcome this shortcoming.

Figure 4: Time evolution of contrail (a) ice crystal number concentration in N cm<sup>-3</sup>, (b) ice water content in mg m<sup>-3</sup>, (c) mean volume radius in μm, and (d) coverage over 4 hours. The Box-and-whisker plots represent observations from COLI at different time points. The black lines represent CASIM-UM model simulation results averaged over all contrails. The shaded areas depict the lower and upper bounds. Observations of contrail age younger than 5 mins are excluded. Biases between model simulations and observations are shown in each panel.

The simulated ice water content (Fig. 4b) increases after initiation and reaches its maximum at the end of the first hour after formation, followed by a decline by a factor of two within four hours due to sedimentation-driven reductions in large ice crystals and the depletion of available water vapour within the contrail cirrus volume. This evolution of ice water content agrees well with observations and previous studies (Lewellen, 2014), with a slight underestimation of the initial contrail ice water content by 1.76 mg m<sup>-3</sup>. The peak in ice water content occurs earlier in our CASIM-UM simulations (i.e. within two

hours) compared to existing ECHAM modelling results (four-six hours) (Bock and Burkhardt, 2016b). The growth of newly formed contrail ice particles depends on the contrail ice number concentration and the amount of water vapour available for deposition. Early in the contrail lifecycle, ice crystal concentrations are sufficiently high to allow almost all available water vapour within the contrail volume to deposit onto the ice crystals. Later, with the decrease in the ice crystal number concentration, the rate of deposition is substantially reduced.






The volume-mean contrail cirrus ice particle radius (Fig. 4c) is calculated based on the equation of spherical shape ice crystals and a young contrail ice density of 200 kg m<sup>-3</sup> in order to ensure consistency between the contrail cirrus parameterisation and the CASIM cloud microphysics scheme. An effective bulk density of 200 kg m<sup>-3</sup> is used in CASIM as it is based on the mass dimension relations from Mitchell (1996) for the ice crystals with sizes of around 70 µm. Here, young-contrail properties are passed to natural clouds shortly after they are calculated by the parameterisation. From that point onward, the subsequent evolution of contrail properties—including cloud ice density—is determined by the model. Therefore, even if a larger value for contrail ice density was initially prescribed, it would only have a minor impact on the subsequent evolution of contrail cirrus. In addition, the chosen value of 200 kg m<sup>-3</sup> is close to the lower end of the observed contrail ice density range of 250-400 kg m<sup>-3</sup> reported in Schumann et al. (2017). We also note that the use of the equation for spherical particles does not necessarily imply a spherical geometry of contrail ice crystals. It merely represents the fact that the ice particles can extend in 3 dimensions, e.g. bullet rosettes. Also, in this case, the density is an effective density of a sphere that encompasses a more complex 3D particle shape. Throughout the contrail lifecycle, the mean radius increases continuously as the ice crystals grow by taking up available water vapour from the surrounding ambient air. This growth is most rapid in the early stages of contrail formation, where high ice supersaturation and abundant water vapour drive significant depositional growth. As the contrail evolves, the growth rate slows due to a decline in ice crystal number concentration and available water vapour within the contrail cirrus volume. The simulated contrail radius is larger than the observed values: in the best estimate simulation it is initialized at 3 um and quickly increases to about 10 um in a few minutes, driven by the fast growth rate of small cloud ice particles in CASIM.

The contrail fraction is calculated using the contrail parameterisation and represents the volume development of contrails over time. As shown in Fig. 4d it follows a similar variation pattern to ice mass (Fig. 4b). Initially, it increases as the contrail cluster expands in width and depth, driven by the contrail ice crystal growth and the plume natural dispersion due to wind shear and sedimentation. Unlike contrail ice mass, which peaks earlier, the contrail fraction reaches its maximum around one hour after formation, as sedimentation contributes to the expansion of contrail volume in the later stages. Eventually, the contrail fraction begins to decline due to ice mass loss from sedimentation and evaporation as it falls into subsaturated air.

Figure 5 shows the vertical evolution of contrail-induced changes in cloud microphysical properties and ambient humidity and temperature since injection. The values are horizontally averaged over the contrail cirrus cluster. At the initial contrail formation levels (five levels in total from 170 to 260 hPa) the ice crystal number concentration (Fig. 5a) peaks at approximately 1 cm<sup>-3</sup> right after formation and then decreases rapidly to less than 0.01 cm<sup>-3</sup>. Below the flight levels, at e.g. 270 hPa, the number concentrations drop to 0.001 cm<sup>-3</sup>, which is nearly an order of magnitude smaller than the results at the flight levels.

This vertical gradient demonstrates the effects of sedimentation, ice crystal size sorting, and the gradual dispersal of ice crystals as contrail evolves.

Contrail ice water content (Fig. 5b) increases rapidly caused by water vapour deposition at the contrail formation level (Fig. 5e). This process is further modulated by sedimentation, which removes larger ice crystals from higher altitudes. Ice crystals grow larger at lower levels, as demonstrated in Fig. 5c, where more water vapour is available, reaching an average size of up to 100 µm before they sediment into subsaturated areas. The continued growth of the mean radius also reflects the evolution of contrail cirrus towards characteristics of natural cirrus clouds, with fewer, larger ice crystals becoming dominant in the later stages of the contrail lifecycle.




Contrail evolution is also closely correlated with feedbacks from ambient humidity (Fig. 5e) and temperature (Fig. 5f). In the beginning, high concentrations of small contrail ice crystals rapidly consume the ambient supersaturated vapour, leading to a decrease near the contrail formation levels. As contrail ice crystal concentrations decline and sedimentation removes ice from these levels, the direct impact on supersaturated vapour diminishes. Meanwhile, at levels below, the sedimentation and sublimation of large ice crystals facilitates the downward transport of water vapour from above, resulting in an increase in vapour. At the contrail formation levels, warming occurs due to latent heat release from contrail ice growth and associated radiative processes. At lower levels, the sedimentation of ice crystals initially contributes to mild warming, which is later offset by cooling from evaporation. The generally good consistency between the simulated contrail evolution and the observations indicates that the model can realistically simulate the evolution of contrail microphysical properties, which is essential for producing reliable contrail cirrus ERF estimates.

Figure 5: Vertical profiles of key in-contrail properties and atmospheric changes over four hours of contrail evolution. In-contrail (a) number concentration (N cm<sup>-3</sup>), (b) ice water content (mg m<sup>-3</sup>), and (c) volume-mean ice crystal radius (μm). Additionally, contrail-induced changes in (d) ice cloud cover (fraction), (e) supersaturation (kg kg<sup>-1</sup>), and (f) temperature (°C) are shown.

## 3.2 Contrail simulation over Europe

We also perform regional simulations with the AEDT air traffic inventory covering the entire domain (35°N-58°N and 10°W-22°E). There are widespread increases in ice cloud cover, ice number concentration, and ice water content induced by the presence of air traffic in the European regional simulations with parameterised contrails (Fig. 6), compared to simulations without parameterised contrails. The spatial distribution of these increases aligns with air traffic volume (Fig. 1), with peak values concentrated in regions of dense air traffic. Note that unlike the contrail cluster experiment conducted under clear-sky conditions, these long-term regional simulations also include feedbacks from the ambient atmosphere and natural clouds.

Zhang et al. (2025) performed contrail cirrus simulations using the UM coupled with a one-moment cloud microphysics scheme (Wilson and Ballard, 1999) (hereafter referred to as one-moment UM) at the global scale. Their results show only a marginal increase in ice water path attributable to contrail cirrus, due to the limitations of the one-moment microphysics approach and the use of a single ice category within one-moment UM. These constraints lead to the representation of contrail ice particles with unrealistically large sizes, resembling those of natural cirrus clouds, thereby affecting microphysical process rates and altering the simulated lifecycle of contrails. Unlike the one-moment cloud microphysics scheme, CASIM, a double-moment cloud microphysics scheme, explicitly predicts ice crystal number concentrations and size distributions. This capability is critical for representing the high number concentrations and small particle sizes characteristic of young contrails. The increases in cloud ice water content simulated by CASIM-UM (Fig. 6a) are substantially larger compared to those in the one-moment UM presented in Zhang et al. (2025; Fig. 6a). In CASIM-UM, increases in ice number concentration (Fig. 6b) can reach up to approximately  $1.5 \times 10^5$  kg<sup>-1</sup> around major air traffic routes. The increase in cloud cover in the UTLS in CASIM-UM is consistent with previous modelling studies and observations (Zhang et al., 2025; Bock and Burkhardt, 2016a). However, it contrasts with CAM results, where cloud cover decreases due to the presence of contrails. Gettelman et al. (2021) attributed this CAM simulated decrease to a drop in relative humidity driven by the temperature increase from the added contrail ice mass.

Figure 6: Spatial distribution of annual mean (averaged over the 10 days in all twelve months) changes in cloud (a) ice water content in kg kg<sup>-1</sup>, (b) ice number concentration in cm<sup>-1</sup>, and (c) ice cloud coverage.

Overall, the increases in cloud cover, ice number concentration, and ice mass extend throughout most of the troposphere (Fig. 7). The most significant increases in ice crystal number concentration and cloud cover occur between 180 and 300 hPa, aligning

with the primary contrail formation levels. Meanwhile, the greatest increases in cloud ice mass appear between 210 and 450 hPa, where sedimentation of large contrail ice particles plays a key role, as observed in the contrail cluster experiment (Section 3.1). Compared to the results in Zhang et al. (2025), CASIM-UM shows more substantial vertical changes in cloud microphysical properties. Zhang et al. (2025) reported a decrease in cloud fraction below the contrail formation altitude in one-moment UM, while CASIM-UM instead simulates an increase.

Figure 7: Vertical cross section of annual mean changes in cloud (a) coverage, (b) ice crystal number concentration (m<sup>-3</sup>), and (c) cloud ice mass (kg m<sup>-3</sup>).

## 3.3 Contrail cirrus effective radiative forcing estimates






The contrail cirrus ERF is determined by calculating the difference between simulations with and without the contrail parameterisation. The annual mean regional contrail cirrus ERF for 2018 over Europe is estimated to be 0.96 W m<sup>-2</sup> from our 'best estimate' CASIM-UM simulations using the best estimate for the initial contrail width, depths, and ice crystal size, as defined in Section 2.3 and Table 1. In contrast to the negligible ERF estimates from one-moment UM in Zhang et al. (2025), the higher ERF of CASIM-UM reflects the ability of a double-moment microphysics scheme to explicitly simulate both contrail ice water content and ice crystal number concentration, resulting in a more realistic representation of the contrail microphysical evolution and more reliable estimates of contrail cirrus ERF. The net ERF (Fig. 8a) is predominantly positive, with stronger signals in regions of denser aviation traffic. The shortwave component (Fig. 8b) is negative, indicating a cooling effect from increased reflection, whereas the longwave component (Fig. 8c) is positive, indicating warming from enhanced infrared radiation trapping. The net ERF is thus governed by the balance of these competing effects, resulting in an overall positive forcing across much of the domain.

Figure 8: Spatial distribution of annual mean (averaged over the 10 days in all twelve months) radiative forcing components over the European domain corresponding to the year of 2018. Contrail cirrus (a) net, (b) shortwave, and (c) longwave ERF.

In addition to the simulations using the 'best estimate' contrail properties, we also perform simulations employing 'upper bound' and 'lower bound' contrail property scenarios. The range of contrail cirrus ERF estimates averaged over this domain spans from 0.19 W m<sup>-2</sup> to 2.80 W m<sup>-2</sup>, based on the 'lower bound' and 'upper bound' CASIM-UM simulations (Table 1), indicating the substantial contribution of the uncertainty in initial contrail width, depth, and ice crystal size on contrail cirrus ERF estimates. A smaller initial ice crystal radius produces higher ice number concentrations for a given ice water content. This increases optical depth by enhancing reflectivity and prolongs radiative impacts, as smaller crystals sediment more slowly and have longer lifetimes. The initial contrail volume also contributes to the ERF range: larger cross sections allow more ambient water vapour to be entrained, thereby increasing the ice water content. As a result, the largest ERF estimates arise from a combination of smaller initial ice crystal radius and larger initial contrail volumes. This upper bound combination produces a factor of ~15 difference in contrail cirrus ERF compared with the lower bound, reflecting the nonlinear dependence of contrail cirrus ERF on the initial ice crystal radius and contrail volume size.

Ambient relative humidity is another critical factor controlling contrail lifetime and ERF. Contrails formed in ice supersaturated regions can persist and spread, while those formed in subsaturated environments dissipate quickly. The magnitude of ice supersaturation also strongly influences contrail ice water content, as it is the main source of depositional water vapour contributing to contrail ice water content, more than the direct engine water vapour emissions (Zhang et al., 2025). Temperature further modulates contrail radiative effects, as colder ambient conditions favour higher ice supersaturation,

contrail formation, and enhanced deposition growth. These sensitivities highlight the need for careful treatment of initial contrail properties and meteorological conditions in assessments of contrail cirrus climate impacts.

The contrail cirrus regional ERF estimate from CAM over this European domain for 2018 is 1.31 W m<sup>-2</sup>, derived by reanalysing the data presented in Zhang et al. (2025). This is much larger than the CASIM-UM lower-bound ERF estimate of 0.19 W m<sup>-2</sup>, obtained using initial contrail properties similar to the CAM simulation (i.e., CAM uses an initial contrail width and depth of 100 m and an ice crystal radius of 3.75 µm). Zhang et al., 2025 has shown that the UM exhibits a more frequent ice supersaturation in the UTLS than CAM, particularly in dense air traffic areas such as Europe. This results in more favourable conditions for contrail formation and persistence in UM simulations (Zhang et al., 2025). Despite this, differences in cloud microphysical, cloud fraction and explicit representation of convection processes between the global and regional models contribute to variations in the hydrological cycle and resulting ERF. In the coarse resolution global UM, convection is parameterised, whereas in the high resolution regional UM, it is explicitly resolved, leading to distinct moisture distributions that consequently alter background humidity and ice supersaturation.

The contrail cirrus ERF simulated by ECHAM over the European domain is approximately 0.15 W m<sup>-2</sup> for 2006 and 0.32 W m<sup>-2</sup> for 2050, as estimated from Fig. 3b of Bock and Burkhardt (2019) and converted from RF to ERF using the factor of 0.42 proposed by Lee et al. (2021). In their configuration, the initial contrail dimensions are 200 m × 200 m, and the number concentration is prescribed at 150 cm<sup>-3</sup> rather than calculated from an assumed ice crystal radius. This setup likely falls between our best-estimate and lower-bound initial contrail properties, which could be attributed to the larger domain extent that covers areas of low air traffic density as well as imply the uncertainty in contrail cirrus ERF estimates due to the host climate models. Overall, the CASIM-UM best estimate of contrail cirrus ERF for the European region falls between the existing CAM and ECHAM estimates. The range defined by our lower and upper bound simulations encompass both, highlighting the model sensitivity to contrail microphysical properties and the associated uncertainties in ERF estimation.

The model resolution also has an impact on the simulated contrail cirrus ERF. While the resolution does not directly affect individual contrail properties (e.g. contrail ice mass mixing ratio, number concentration, and fraction), it indirectly affects contrails by modifying meteorological conditions, underlying physical processes, and cloud overlap. Vertical and horizontal resolutions of climate models affect their representation of ice supersaturation, even when using the same underlying model framework. At coarser resolutions, small-scale processes such as localized motions, wave-driven dynamics, and turbulence are either poorly resolved or entirely smoothed out, leading to a less realistic representation of the variability in upper tropospheric humidity. This often results in an underestimation of ice supersaturation, as the conditions necessary for its formation—localized cooling and saturation anomalies—are inadequately captured. These differences in the resolved meteorology can in turn affect contrail formation, persistence, and radiative impacts. In addition, vertical resolution can influence the simulated contrail cirrus ERF through its effect on the representation of cloud overlap and the vertical cloud structure, which alters radiative transfer calculations (Chen et al., 2013). To assess the impact of horizontal resolution on contrail cirrus ERF, idealised CASIM-UM runs are performed at varying horizontal resolutions with fixed total flight distance over the region. Our results indicate that as the resolution becomes coarser, the contrail cirrus ERF decreases. From the finest

resolution of ~4 km to the coarsest resolution of ~40 km, the contrail cirrus ERF is reduced by 6.4%. However, the domain-averaged ERF values remain within 10% of each other, suggesting that the overall impact on contrail cirrus ERF of the horizontal resolutions within this range (i.e. 4-40 km) is relatively modest.



The simulated contrail cirrus ERF exhibits a distinct seasonal cycle (Fig. 9a), reflecting the combined influence of background cloudiness and meteorology (Fig. 9c), insolation, and air traffic volume (Fig. 9b). The net ERF peaks in winter (November–January), when shortwave cooling is weakest, and approaches zero from spring to early autumn (March–September), when shortwave cooling and longwave warming are of comparable magnitudes. Previous studies have attributed the seasonal cycle of contrail cirrus ERF to variations in contrail formation and persistence, with contrails forming and persisting less frequently in summer and more frequently in winter due to changes in meteorological conditions (Chen et al., 2013; Bock and Burkhardt, 2016a). In our simulations, however, contrail cirrus coverage remains relatively modest (~0.10–0.15). However, we note that contrail cirrus coverage is defined here as the change in total cloud fraction, thereby reflecting not only the direct contrail contribution but also the changes in natural cloudiness induced by contrails. Contrail cirrus coverage is lowest in summer, despite the pronounced summer maximum in flight distance, highlighting that contrail occurrence and persistence are primarily constrained by temperature and humidity conditions.

The seasonal cycle of total cloud fraction (Fig. 9c) is found to be closely linked to the contrail cirrus ERF. Contrails primarily enhance upper-tropospheric cloudiness between 200–300 hPa (Fig. 9d), with the largest increases in spring and late autumn to winter, coinciding with the longwave ERF peaks. By contrast, the shortwave ERF is weakest in autumn and winter (October–February). This is partly due to a relatively high amount of background low-level clouds, which leads to a 'cloud masking' effect of the additional shortwave reflection due to contrail cirrus and thereby weakens the contrail-induced shortwave cooling.

Figure 9: Annual cycles of contrail cirrus median net, shortwave, and longwave ERFs (with 10–90% ranges of daily values) (a), contrail cirrus coverage and air traffic distance flown (b), vertical distribution of area-averaged total cloud fraction in the simulations without aviation (c), and vertical distribution of area-averaged contrail-induced changes in total cloud fraction (d).

#### 4 Conclusions




This study investigates the contrail cirrus evolution and radiative impacts using a newly implemented contrail scheme in the CASIM double-moment cloud microphysics scheme within the UM climate model. Using a contrail cluster experiment and a set of regional simulations over a European domain, we show that CASIM-UM is able to effectively represent the contrail cirrus evolution, therefore providing independent estimates of contrail cirrus ERF.

In the contrail cluster experiment, the evolution of contrail microphysical properties was analysed under clear sky and ice supersaturated ambient conditions. The main characteristics of contrail evolution in CASIM-UM align well with observations (Schumann et al., 2017) and previous modelling studies (Bock and Burkhardt, 2016b). CASIM-UM produces large initial ice-crystal number concentrations within young contrails, followed by a gradual decline over time. The simulated ice-water content peaks at the end of the first hour and then decreases progressively. The vertical evolution of contrail properties highlights the significant role of sedimentation in contrail development, particularly in redistributing ice mass and number concentration across different atmospheric levels. Other key features of contrail cirrus evolution, including the growth of the ice crystal radius, the expansion of contrail fraction, and the feedbacks of ambient humidity and temperature over time, are also well captured in the CASIM-UM model experiments. However, we note that the simulated initial contrail ice number concentration and water content are underestimated. This resulted from an overestimation of the initial contrail fraction, which is diagnosed by the bimodal cloud scheme under the clear-sky conditions of the experiment. In addition, the simulated contrail radius exceeds observations, indicating a fast growth rate of small cloud ice in CASIM.

Our regional CASIM-UM simulations over Europe demonstrate the regional impact of contrail cirrus on cloud properties and the radiation budget. There is a widespread increase in the ice cloud cover, ice number concentration, and ice water content across the domain. Compared to previous studies using the UM with a one-moment microphysics scheme, the double-moment scheme in CASIM-UM produces a larger increase in cloud-ice number concentrations and ice water content, leading to a more realistic representation of contrail-induced cloud modifications.

The annual mean ERF over Europe for 2018 is estimated to be 0.93 W m<sup>-2</sup>, with a range between 0.19 and 2.80 W m<sup>-2</sup> when accounting for the observed variation in initial contrail width, depth, and ice crystal size. This large range highlights the uncertainty in contrail-cirrus ERF arising from the observational variability in these initial contrail characteristics. The contrail cirrus ERF estimates from previous studies with CAM (Zhang et al., 2025) and ECHAM (Bock and Burkhardt, 2016a) also fall within this range. The simulated contrail cirrus ERF exhibits a distinct seasonal cycle, peaking in winter when shortwave cooling is weakest and approaching zero in summer when longwave warming and shortwave cooling offset each other. Our analysis also highlights the dominant role of meteorology and background clouds in controlling contrail cirrus occurrence and its radiative impact.

While our use of a regional European domain nested within a global UM framework enables high-resolution simulation of contrail microphysics, it is important to acknowledge the inherent limitations of such a setup. The hourly boundary conditions derived from the global model constrain regional variability and may introduce biases in ice supersaturation and synoptic meteorology. Notably, the absence of two-way coupling means that contrail-induced perturbations in radiation, humidity, or cloudiness cannot influence the host global circulation, potentially limiting the broader response that would emerge under a truly global coupled system. In addition, our European domain ERF estimate may not directly extrapolate to other regions or to globally integrated metrics. Differences in background climate, aviation density, ice supersaturation frequency, and meteorological conditions elsewhere could result in substantially different contrail ERF.

This study represents a significant step forward in contrail modelling within the UM and provides insights for future assessment of contrail cirrus climate impact. Our results highlight the importance of accurately representing the key contrail feature of high ice number concentration in contrail-cirrus climate simulations. Future research should involve: (1) extending simulations to other domains with dense air traffic density to enhance the understanding of global contrail cirrus effects; (2) investigating the influence of alternative aviation fuels on contrail properties, particularly their impacts on ice crystal number concentrations and contrail lifetime; and (3) further assessment of uncertainties in contrail cirrus ERF estimates from the representation of ice supersaturation in climate models.

## Data availability

The UM code and its configuration files are subject to Crown Copyright. A licence for the UM can be requested from <a href="https://www.metoffice.gov.uk/research/approach/collaboration/unified-model/partnership">https://www.metoffice.gov.uk/research/approach/collaboration/unified-model/partnership</a>. Data for reproducing the figures in this paper will be accessible via Zenodo.

#### 480 Author contributions

WZ: Implementation of the contrail parameterisation in the CASIM-UM, CASIM-UM contrail cirrus modelling, data analysis, writing (original draft preparation and editing), and conceptualisation. PRF: Support with implementation of the contrail parameterisation in the CASIM-UM and CASIM-UM contrail cirrus modelling and writing (review and editing). KVW, CJM, and PMF: writing (review and editing). AR: Writing (review and editing) and conceptualisation.

#### 485 Competing interests

The contact author has declared that none of the authors has any competing interests

#### Acknowledgements

490

Weiyu Zhang is supported by the Leeds-York-Hull Natural Environment Research Council (NERC) Doctoral Training Partnership (DTP) Panorama under grant NE/S007458/1 with the UK Met Office CASE partnership. Alexandru Rap and Piers Forster acknowledge supports from the EPSRC TOZCA grant (EP/V000772/1) and the NERC MAGICA grant (NE/Z503836/1). We acknowledge use of the Monsoon2 system for UM simulations, a collaborative facility supplied under the Joint Weather and Climate Research Programme, a strategic partnership between the Met Office and NERC.

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
