# Peer review of "Modelling contrail cirrus using a double-moment cloud microphysics scheme in the UK Met Office Unified Model"

_EGUsphere, 2025_

## Author Comment (AC1)

We appreciate the time and effort that Reviewer 1 has taken to review our manuscript and we thank them for their useful comments and suggestions on improving the paper. We have now addressed all the comments and made the necessary revisions to the manuscript. Please see our point-by-point responses below.

**Suggestions for Improvement**

**While the manuscript mentions comparison with observations and other models, more quantitative evaluation of the model's performance (e.g., ice water content, number concentration, cloud fraction) against available satellite or in-situ observations would strengthen the study.**

**If possible, include statistical metrics (e.g., bias) for key variables.**

Thank you for this suggestion. We have now included the calculated bias of the simulated means of in-contrail ice number concentration, ice water content, and volume mean radius with respect to the observation medians in Fig. 4 and updated the discussion of the idealised contrail cluster experiment analysis in the revised manuscript as follows:

L 216-217: *'…, although the simulated initial in-contrail number concentration is smaller than observed by 0.65 cm$^{-3}$ on average.'*

L 238-240: *'This evolution of ice water content agrees **well with observations and previous studies (Lewellen, 2014), with a slight underestimation of the initial contrail ice water content by 1.76 mg m$^{-3}$.'***

[Figure]

**Figure 4: Time evolution of contrail (a) ice crystal number concentration in N cm$^{-3}$, (b) ice water content in mg m$^{-3}$, (c) mean volume radius in μm, and (d) coverage over 4 hours. The Box-and-whisker plots represent observations from COLI at different time points. The black lines represent CASIM-UM model simulation results averaged over all contrails. The shaded areas depict the lower and upper bounds. Observations of contrail age younger than 5 mins are excluded. Biases between model simulations and observations are shown in each panel.**

**The manuscript discusses uncertainties in contrail ERF, but does not provide a detailed sensitivity analysis of the model's key parameters (e.g., initial ice crystal size, number concentration, ambient humidity).**

**Consider adding a section or at least a discussion on how sensitive your results are to these parameters.**

To address this point we have now extended the discussion in order to:

1) clarify how the lower and upper bounds of contrail ERF reflect sensitivities to initial ice crystal radius and cross section size.
2) discuss the critical role of ambient humidity and temperature in controlling contrail formation, properties and ERF.

as follows in L 355- 372:

*'In addition to the simulations using the 'best estimate' contrail properties, we also perform simulations employing 'upper bound' and 'lower bound' contrail property scenarios. The range of contrail cirrus ERF estimates averaged over this domain spans from 0.19 W m$^{-2}$ to 2.80 W m$^{-2}$, based on the 'lower bound' and 'upper bound' CASIM-UM simulations (Table 1), indicating the substantial contribution of the uncertainty in initial contrail width, depth, and ice crystal size on contrail cirrus ERF estimates. A smaller initial ice crystal radius produces higher ice number concentrations for a given ice water content. This increases optical depth by enhancing reflectivity and prolongs radiative impacts, as smaller crystals sediment more slowly and have longer lifetimes. The initial contrail volume also contributes to the ERF range: larger cross sections allow more ambient water vapour to be entrained, thereby increasing the ice water content. As a result, the largest ERF estimates arise from a combination of smaller initial ice crystal radius and larger initial contrail volumes. This upper bound combination produces a factor of ~15 difference in contrail cirrus ERF compared with the lower bound, reflecting the nonlinear dependence of contrail cirrus ERF on the initial ice crystal radius and contrail volume size.*

*Ambient relative humidity is another critical factor controlling contrail lifetime and ERF. Contrails formed in ice supersaturated regions can persist and spread, while those formed in subsaturated environments dissipate quickly. The magnitude of ice supersaturation also strongly influences contrail ice water content, as it is the main source of depositional water vapour contributing to contrail ice water content, substantially more than the direct engine water vapour emissions (Zhang et al., 2025). Temperature further modulates contrail radiative effects, as colder ambient conditions favour higher ice supersaturation, contrail formation, and enhanced deposition growth. These sensitivities highlight the need for careful treatment of initial contrail properties and meteorological conditions in assessments of contrail cirrus climate impacts.'*

**Please clarify the rationale for selecting 70 vertical levels and the vertical resolution profile, especially in the context of representing contrail processes in the UTLS.**

**Indicate how the chosen vertical and horizontal resolutions impact the simulation of contrail lifecycles and radiative effects.**

The model configuration with 70 vertical levels follows the standard RAL2 (Bush et al. 2022) setup of the UM, providing a vertical resolution of about 500 m in the UTLS, which is comparable to typical scales used in global climate models for representing UTLS processes (e.g. Chen et al. (2012), Gettelman et al. (2021), and Zhang et al. (2025)).

To further address this point, we have now extended the discussion on the sensitivity of contrail cirrus ERF to model resolution, also discussing the associated impacts on contrail lifecycles and radiative effects as follows in L 392-407:

*'The model resolution also has an impact on the simulated contrail cirrus ERF. While the resolution does not directly affect individual contrail properties (e.g. contrail ice mass mixing ratio, number concentration, and fraction), it indirectly affects contrails by modifying meteorological conditions, underlying physical processes, and cloud overlap. Vertical and horizontal resolutions of climate models affect their representation of ice supersaturation, even when using the same underlying model framework. At coarser resolutions, small-scale processes such as localized motions, wave-driven dynamics, and turbulence are either poorly resolved or entirely smoothed out, leading to a less realistic representation of the variability in upper tropospheric humidity. This often results in an underestimation of ice supersaturation, as the conditions necessary for its formation—localized cooling and saturation anomalies—are inadequately captured. These differences in the resolved meteorology can in turn affect contrail formation, persistence, and radiative impacts. In addition, vertical resolution can influence the simulated contrail cirrus ERF through its effect on the representation of cloud overlap and the vertical cloud structure, which alters radiative transfer calculations (Chen et al., 2013). To assess the impact of horizontal resolution on contrail cirrus ERF, idealised CASIM-UM runs are performed at varying horizontal resolutions with fixed total flight distance over the region. Our results indicate that as the resolution becomes coarser, the contrail cirrus ERF decreases. From the finest resolution of ~4 km to the coarsest resolution of ~40 km, the contrail cirrus ERF is reduced by 6.4%. However, the domain-averaged ERF values remain within 10% of each other, suggesting that the overall impact on contrail cirrus ERF of the horizontal resolutions within this range (i.e. 4-40 km) is relatively modest.'*

**The discussion could be expanded to address the limitations of regional modeling (e.g., boundary effects, lack of global feedbacks) and how these might affect the generalizability of the results. Consider elaborating on the implications for global-scale modeling and policy.**

We agree that it is important to clarify this. We have now extended the discussion as follows in L 461-468:

*'While our use of a regional European domain nested within a global UM framework enables high-resolution simulation of contrail microphysics, it is important to acknowledge the inherent limitations of this setup. The hourly boundary conditions derived from the global model constrain regional variability and may introduce biases in ice supersaturation and synoptic meteorology. Notably, the absence of two-way coupling means that contrail-induced perturbations in radiation, humidity, or cloudiness cannot influence the host global circulation, potentially limiting the broader response that would emerge under a truly global coupled system. In addition, our European domain ERF estimate may not directly extrapolate to other regions or to globally integrated metrics. Differences in*

*background climate, aviation density, ice supersaturation frequency, and meteorological conditions elsewhere could result in substantially different contrail ERF.'*

**Ensure that all figures are clear, with appropriate legends and axis labels.**

We have now reviewed and revised all figures, adding/standardizing legends and axis labels.

**Provide time series or spatial maps of key variables (e.g., contrail coverage, ERF) to illustrate model behavior.**

Thank you for this suggestion. We have now added a substantial new analysis of the annual cycle of the contrail cirrus coverage and ERF to illustrate the model's behaviour. The following new figures and discussion have now been added at L 408-423:

*'The simulated contrail cirrus ERF exhibits a distinct seasonal cycle (Fig. 9a), reflecting the combined influence of background cloudiness and meteorology (Fig. 9c), insolation, and air traffic volume (Fig. 9b). The net ERF peaks in winter (November–January), when shortwave cooling is weakest, and approaches zero from spring to early autumn (March–September), when shortwave cooling and longwave warming are of comparable magnitudes. Previous studies have attributed the seasonal cycle of contrail cirrus ERF to variations in contrail formation and persistence, with contrails forming and persisting less frequently in summer and more frequently in winter due to changes in meteorological conditions (Chen et al., 2013; Bock and Burkhardt, 2016a). In our simulations, however, contrail cirrus coverage remains relatively modest (~0.10–0.15). However, we note that contrail cirrus coverage is defined here as the change in total cloud fraction, thereby reflecting not only the direct contrail contribution but also the changes in natural cloudiness induced by contrails. Contrail cirrus coverage is lowest in summer, despite the pronounced summer maximum in flight distance, highlighting that contrail occurrence and persistence are primarily constrained by temperature and humidity conditions.*

*The seasonal cycle of total cloud fraction (Fig. 9c) is found to be closely linked to the contrail cirrus ERF. Contrails primarily enhance upper-tropospheric cloudiness between 200–300 hPa (Fig. 9d), with the largest increases in spring and late autumn to winter, coinciding with the longwave ERF peaks. By contrast, the shortwave ERF is weakest in autumn and winter (October–February). This is partly due to a relatively high amount of background low-level clouds, which leads to a 'cloud masking' effect of the additional shortwave reflection due to contrail cirrus and thereby weakens the contrail-induced shortwave cooling.'*

In abstract:

*'The seasonal cycle of contrail cirrus ERF is mainly driven by the background meteorology and the natural clouds vertical structure.'*

In Conclusion:

*'The simulated contrail cirrus ERF exhibits a distinct seasonal cycle, peaking in winter when shortwave cooling is weakest and approaching zero in summer when longwave warming and shortwave cooling offset each other. Our analysis also highlights the dominant role of meteorology and background clouds in controlling contrail cirrus occurrence and its radiative impact.'*

[Figure]

**Figure 9: Annual cycles of contrail cirrus median net, shortwave, and longwave ERFs (with 10–90% ranges of daily values) (a), contrail cirrus coverage and air traffic distance flown (b), vertical distribution of area-averaged total cloud fraction in the simulations without aviation (c), and vertical distribution of area-averaged contrail-induced changes in total cloud fraction (d).**

**Minor comments**

**Some references to previous studies could be updated or expanded, particularly regarding recent advances in contrail observation and modeling.**

We have now updated the introduction section to include references to more recent studies on contrail observation and modelling:

- *Gruber, S., Unterstrasser, S., Bechtold, J., Vogel, H., Jung, M., Pak, H., and Vogel, B.: Contrails and their impact on shortwave radiation and photovoltaic power production – a regional model study, Atmos. Chem. Phys., 18, 6393–6411, https://doi.org/10.5194/acp-18-6393-2018, 2018.*
- *Lottermoser, A. and Unterstrasser, S.: High-resolution modeling of early contrail evolution from hydrogen-powered aircraft, Atmos. Chem. Phys., 25, 7903–7924, https://doi.org/10.5194/acp-25-7903-2025, 2025.*
- *Singh, D. K., Sanyal, S., and Wuebbles, D. J.: Understanding the role of contrails and contrail cirrus in climate change: a global perspective, Atmos. Chem. Phys., 24, 9219–9262, https://doi.org/10.5194/acp-24-9219-2024, 2024.*

- *Wang, Z., Bugliaro, L., Jurkat-Witschas, T., Heller, R., Burkhardt, U., Ziereis, H., Dekoutsidis, G., Wirth, M., Groß, S., Kirschler, S., Kaufmann, S., and Voigt, C.: Observations of microphysical properties and radiative effects of a contrail cirrus outbreak over the North Atlantic, Atmos. Chem. Phys., 23, 1941–1961, https://doi.org/10.5194/acp-23-1941-2023, 2023.*

**Double-check for typographical errors and ensure consistency in units and notation throughout the manuscript.**

We have now reviewed and revised figures.

**For Figure 2, I recommend providing the linear regression equation, including both the intercept and slope, either in the figure panel or the caption. Including this information would enhance the quantitative interpretation of the relationship shown, and allow readers to compare your results with those from other studies more easily. This is a common practice in the field and would further strengthen the clarity and reproducibility of your analysis.**

Thank you for this suggestion. We have now added the linear regression equations in the panels.

[Figure]

**While the introduction provides a solid overview of the current state of contrail cirrus research and cites several recent studies, I recommend including a reference to the recent review by Singh et al. (2024). This comprehensive review synthesizes the latest developments and ongoing challenges in contrail modeling and climate impacts. It would further strengthen the background section by providing readers with a broader context and up-to-date summary of the field.**

We have now cited Singh et al. (2024) in the revised manuscript.

**References**

Bush, M., Boutle, I., Edwards, J., Finnenkoetter, A., Franklin, C., Hanley, K., Jayakumar, A., Lewis, H., Lock, A., Mittermaier, M., Mohandas, S., North, R., Porson, A., Roux, B., Webster, S., and Weeks, M.: The second Met Office Unified Model–JULES Regional Atmosphere and Land configuration, RAL2, Geosci. Model Dev., 16, 1713–1734, https://doi.org/10.5194/gmd-16-1713-2023, 2023.

Chen, C. C., Gettelman, A., Craig, C., Minnis, P., and Duda, D. P.: Global contrail coverage simulated by CAM5 with the inventory of 2006 global aircraft emissions, J. Adv. Model. Earth Syst., 4, https://doi.org/10.1029/2011MS000105, 2012.

Gettelman, A., Chen, C. C., and Bardeen, C. G.: The Climate Impact of COVID19 Induced Contrail Changes, Atmos. Chem. Phys., 2021, 1-17, 10.5194/acp-2021-210, 2021.

Zhang, W., Van Weverberg, K., Morcrette, C. J., Feng, W., Furtado, K., Field, P. R., Chen, C. C., Gettelman, A., Forster, P. M., Marsh, D. R., and Rap, A.: Impact of host climate model on contrail cirrus effective radiative forcing estimates, Atmos. Chem. Phys., 25, 473-489, 10.5194/acp-25-473-2025, 2025.

---

## Author Comment (AC2)

We appreciate the time and effort that Reviewer 2 has taken to review our manuscript and we thank them for their useful comments and suggestions on improving the paper. We have now addressed all the comments and made the necessary revisions to the manuscript. Please see our point-by-point responses below.

**As a general comment, it would be good if the manuscript could present some temporal evolution and variation of the contrail radiative forcing. You have fine scale temporal detail, and you show the evaluation of the 'idealized' case, but I would like to see some use of temporal variation of the longer runs: a diurnal cycle if possible but at least an annual cycle of contrail properties and radiative forcing.**

We thank the reviewer for this very good suggestion. We have now added a new analysis and figure of the annual cycle for contrail cirrus coverage and ERF to illustrate the model's behaviour. Contrail cirrus coverage is derived from the difference in total cloud cover between simulations with and without contrails, because contrail cirrus cover is not in the model output.

The following new figures and discussion have now been added at L 408-423:

'*The simulated contrail cirrus ERF exhibits a distinct seasonal cycle (Fig. 9a), reflecting the combined influence of background cloudiness and meteorology (Fig. 9c), insolation, and air traffic volume (Fig. 9b). The net ERF peaks in winter (November–January), when shortwave cooling is weakest, and approaches zero from spring to early autumn (March–September), when shortwave cooling and longwave warming are of comparable magnitudes. Previous studies have attributed the seasonal cycle of contrail cirrus ERF to variations in contrail formation and persistence, with contrails forming and persisting less frequently in summer and more frequently in winter due to changes in meteorological conditions (Chen et al., 2013; Bock and Burkhardt, 2016a). In our simulations, however, contrail cirrus coverage remains relatively modest (~0.10–0.15). However, we note that contrail cirrus coverage is defined here as the change in total cloud fraction, thereby reflecting not only the direct contrail contribution but also the changes in natural cloudiness induced by contrails. Contrail cirrus coverage is lowest in summer, despite the pronounced summer maximum in flight distance, highlighting that contrail occurrence and persistence are primarily constrained by temperature and humidity conditions.*

*The seasonal cycle of total cloud fraction (Fig. 9c) is found to be closely linked to the contrail cirrus ERF. Contrails primarily enhance upper-tropospheric cloudiness between 200–300 hPa (Fig. 9d), with the largest increases in spring and late autumn to winter, coinciding with the longwave ERF peaks. By contrast, the shortwave ERF is weakest in autumn and winter (October–February). This is partly due to a relatively high amount of background low-level clouds, which leads to a 'cloud masking' effect of the additional shortwave reflection due to contrail cirrus and thereby weakens the contrail-induced shortwave cooling.*'

In abstract:

'*The seasonal cycle of contrail cirrus ERF is mainly driven by the background meteorology and the natural clouds vertical structure.*'

In Conclusion:

*'The simulated contrail cirrus ERF exhibits a distinct seasonal cycle, peaking in winter when shortwave cooling is weakest and approaching zero in summer when longwave warming and shortwave cooling offset each other. Our analysis also highlights the dominant role of meteorology and background clouds in controlling contrail cirrus occurrence and its radiative impact.'*

[Figure]

**Figure 9: Annual cycles of contrail cirrus median net, shortwave, and longwave ERFs (with 10–90% ranges of daily values) (a), contrail cirrus coverage and air traffic distance flown (b), vertical distribution of area-averaged total cloud fraction in the simulations without aviation (c), and vertical distribution of area-averaged contrail-induced changes in total cloud fraction (d).**

**Specific Comments:**

**Page 4, L118: what is the depositional water in the volume? Everything above supersaturation? I guess that's what in equation 4?**

Yes, that is correct. We have now clarified it as follows in the revised manuscript:

*'The ice mass is calculated from both aircraft engine-emitted water vapour and depositional water (the excess water vapour above ice supersaturation) within the contrail volume.'*

**Page 8, L213: does that mean the area is too large?**

The overestimation of contrail fraction is due to the way cloud ice is represented and cloud fraction is diagnosed in the model. We have now clarified it in the revised manuscript in L 219-227:

'*In our study, contrail fraction is defined as the difference in total cloud fraction between a simulation with contrails and one without, and it is treated in the same way as natural cloud ice. In CASIM, contrail fraction is diagnosed using the bimodal cloud fraction scheme (Van Weverberg et al., 2021), which follows the approach of Abel et al. (2017) to diagnose the ice cloud fraction. It incorporates the additional contrail ice into the grid-box subgrid distribution of humidity. As a result, contrail moisture is mixed throughout the grid-box after a single time step—much faster than in reality—leading to lower in-contrail values than observed. In addition, the conversion of cloud ice mass into cloud fraction depends on the assumed mass distribution. In this work, the initial contrail volume width is about 200 m, much smaller than the grid-box area (4 km by 4 km), introducing an overestimation to the initial contrail fraction. Higher-resolution simulations with horizontal grid spacing of around 200 m would be required to overcome this shortcoming.*'

**Page 8, L216: is the cloud fraction magnitude expected just the contrail width? At 200m in a 4km grid box it;s 0.05 right?**

Thank you for pointing out the need for clarification. The value is not just 0.05 in this case, as this represents only the geometric ratio of contrail width to grid-box size and neglects the contrail length. We have now edited the discussion on the initial contrail fraction to avoid potential confusions.

To clarify, in our simulations, contrail ice is added as increments to the natural cloud field and treated in the same way as background natural cloud ice. As a result, the initial contrail cirrus fraction may be overestimated due to the way cloud ice is represented and cloud fraction is diagnosed in the model. Specifically, the overestimation arises from:

1)  Rapid mixing of contrail moisture into the grid-box subgrid humidity distribution.
2)  The conversion of cloud ice mass into cloud fraction with the assumed mass distribution.

To illustrate point (2), we provide the plot below. Although the contrail cirrus ice water content is only of the order of $10^{-6}$ kg kg$^{-1}$ (See Fig. 4b or Fig. 5b), even such small values correspond to a steep gradient in the diagnosed cloud fraction (as seen near $10^{-6}$ on the $\log_{10}$ ice mass axis). This indicates that small increments of contrail ice can disproportionately increase the diagnosed cloud fraction, thereby artificially amplifying cloud cover and leading to an apparent overestimation of contrail cirrus coverage.

[Figure]

Figure 1. Relationship between cloud fraction and cloud water content shown on a log$_{10}$ scale.

We have now update it in the revised manuscript in L 219-227:

'*In our study, contrail fraction is defined as the difference in total cloud fraction between a simulation with contrails and one without, and it is treated in the same way as natural cloud ice. In CASIM, contrail fraction is diagnosed using the bimodal cloud fraction scheme (Van Weverberg et al., 2021), which follows the approach of Abel et al. (2017) to diagnose the ice cloud fraction. It incorporates the additional contrail ice into the grid-box subgrid distribution of humidity. As a result, contrail moisture is mixed throughout the grid-box after a single time step—much faster than in reality—leading to lower in-contrail values than observed. In addition, the conversion of cloud ice mass into cloud fraction depends on the assumed mass distribution. In this work, the initial contrail volume width is about 200 m, much smaller than the grid-box area (4 km by 4 km), introducing an overestimation to the initial contrail fraction. Higher-resolution simulations with horizontal grid spacing of around 200 m would be required to overcome this shortcoming.*'

**Page 9, L220: What if you just assigned the increase in cloud fraction when the contrail is initialized based on the initial width?**

As mentioned above, this is not possible within the methodology of this study since the cloud fraction is diagnosed by the model based on the bimodal cloud fraction scheme (Van Weverberg et al., 2021) and cannot be directly assigned.

**Page 10, L244: does this imply the density of small ice in CASIM is too low? 200 kg m-3 seems very small for small ice crystals….**

We have now clarified the reason for choosing 200 kg m$^{-3}$ for contrail ice density in the revised manuscript in L 246-254:

'*The volume-mean contrail cirrus ice particle radius (Fig. 4c) is calculated based on the equation of spherical shape ice crystals and a young contrail ice density of 200 kg m$^{-3}$ in order*

*to ensure consistency between the contrail cirrus parameterisation and the CASIM cloud microphysics scheme. An effective bulk density of 200 kg m⁻³ is used in CASIM as it is based on the mass dimension relations from Mitchell (1996) for the ice crystals with sizes of around 70 μm. Here, young-contrail properties are passed to natural clouds shortly after they are calculated by the parameterisation. From that point onward, the subsequent evolution of contrail properties—including cloud ice density—is determined by the model. Therefore, even if a larger value for contrail ice density was initially prescribed, it would only have a minor impact on the subsequent evolution of contrail cirrus. In addition, the chosen value of 200 kg m⁻³ is close to the lower end of the observed contrail ice density range of 250–400 kg m⁻³ reported in Schumann et al. (2017).'*

**Page 10, L264: define QCF in figure 5b title. Is that the model variable for ice water content?**

Yes, that's right - thank you for pointing it out. We have now replaced 'QCF' with 'ice water content' in the figure title.

**Page 10, L270: is figure 5e just change in water vapor? Or is it change in mass above supersaturation? The latter is a function of temperature change as well, so not as clear. Suggest better to show just r water vapor change.**

Figure 5e shows the change in excess water vapour above ice saturation (Δ(r−r_icesat); Fig. 2a below). To address the temperature-dependence concern, we made a plot for specific humidity change (Δr; Fig. 2b) as suggested by the reviewer.

The two fields are almost indistinguishable as a residual plot (Fig. 2c) is near zero. This reflects the small temperature perturbations from the single contrail cluster (as the temperature change shown in Fig. 5f in the manuscript) and the proximity to ice saturation (r_icesat).

To avoid redundancy, we prefer to retain the original Fig. 5e.

[Figure]

Figure 2. (a) Change in supersaturated water (kg kg⁻¹), (b) change in specific humidity (kg kg⁻¹), and (c) residual (a)–(b).

**Page 12, L290: is figure 6 all 12 x 10-day periods averaged? Please state what time range is being shown.**

Yes, that's correct. We have now clarified it in the figure caption as follows:

'*Figure 6: Spatial distribution of annual mean (averaged over the 10 days in all twelve months) changes in cloud (a) ice water content in kg kg$^{-1}$, (b) ice number concentration in kg$^{-1}$, and (c) ice cloud coverage.*'

**Page 12, L304: This seems high given that it is near the peak of the contrail # concentration in figure 5a after 30 min.**

Sorry for the confusion and thank you for pointing this out. Ice number concentration was in different units in these two figures. Fig. 6 presented the direct output from the model in units of N kg$^{-1}$, whereas Figure 5 (a) shows number concentration converted to N cm$^{-3}$ (i.e. by multiplying with air density) for consistency with the units used in the in situ observations.

To avoid this confusion, we have now converted the number concentration in Fig. 6 (b) from N kg$^{-1}$ to N cm$^{-3}$ in the revised manuscript to allow direct comparison with Fig. 5 (a). The values in Fig. 6(b) are now more comparable to Fig. 5 (a).

**Page 14, L340: Figure 8. It would be useful to note what time period this represents. An annual average? What does the annual cycle look like?**

We have now clarified the time period in the caption as: '*Figure 8: Spatial distribution of annual mean (averaged over the 10 days in all twelve months) radiative forcing components …*'.

As mentioned in our response to the general comment, we have now added the annual cycle of contrail cirrus ERF in the revised manuscript.

**Page 15, L379: what about the annual or diurnal cycle of contrail forcing? What frequency of AEDT inventory did you use (e.g. you can get it hourly). I assume at least monthly variation to do an annual cycle.**

As we used the monthly AEDT inventory in this work, we cannot provide a full analysis of the diurnal cycle. However, as mentioned in our response to the general comment, we have now added an analysis and figure of the annual cycle of contrail cirrus ERF in the revised manuscript.

---

## Author Response (AR2)

1. P2, L46-47. "often exceeding 104 cm-3 compared to typical natural cirrus values of 10-100 cm-3". Please double check the two numbers given here. Based on Figure 1 of Schumann et al., Nice is generally below 104 cm-3. Are you sure typical natural cirrus values of 10-100 cm-3? These numbers contrast those you show in Fig. 4 of this manuscript.

Thank you for catching this error. This was due to a unit mix-up. We now use consistent cm-3 values and revised in the manuscript as follows:

'often exceeding 10-2 cm-3 compared to typical natural cirrus values of 10-4-10-2 cm-3

2. P2, L56-58. The year citation format differs from other places.

Thank you for pointing this out, we have now corrected it in the revised manuscript:

'(e.g. Lewellen, 2014; Unterstrasser, 2016; Schumann, 2012; Lottermoser and Unterstrasser, 2025), while numerical weather prediction and climate models capture large-scale properties and rapid atmospheric adjustments (e.g. Chen and Gettelman, 2013; Bock and Burkhardt, 2016a)'

3. Figure 3. X-axis and caption: Use unit of percentage for RHice to be consistent with the discussion in the text and general tradition in the field.

As suggested, we have now changed the units of the X-axis in Fig. 3 from fraction to "%".

4. P13, L316. Unit inconsistency with the revised Fig 6b. Please double check other places to ensure consistency.

Thank you for pointing this out. We have now changed to a consistent use of cm-3 in the revised manuscript, rather than kg-1.

5. Fig6b caption. A typo in the unit.

Thank you. We have now corrected it from cm-1 to cm-3 in the revised manuscript.

6. Data availability. "Data for reproducing the figures in this paper will be accessible via Zenodo." Please update this with a link to Zenodo.

**This has now been updated as:**

'AEDT inventory for reproducing Fig. 1 is available upon request at <a href="https://aedt.faa.gov/">https://aedt.faa.gov/</a>. COLI dataset for reproducing Fig. 2 is available in the supplement of Schumann et al. (2017) (<a href="https://acp.copernicus.org/articles/17/403/2017/">https://acp.copernicus.org/articles/17/403/2017/</a>). Data for all remaining figures are available at <a href="https://doi.org/10.5281/zenodo.17203286">https://doi.org/10.5281/zenodo.17203286</a>.'

We have updated Fig. 7b to convert the unit to cm-3 for consistency with the other figures in the manuscript. We also corrected typos in the caption of Fig. 7.

We have now also added in Acknowledgements a sentence to thank the editor and the two reviewers for their careful revision and useful suggestions.